Higher body condition with infection by Haemoproteus parasites in Bananaquits (Coereba flaveola)

Gutiérrez-Ramos Nicole A. 1 nicole.gutierrez1@upr.edu
http://orcid.org/0000-0002-8289-1497 Acevedo Miguel A. 2
1 Department of Biology, Universidad de Puerto Rico, Recinto de Rio Piedras , San Juan , Puerto Rico
2 Department of Wildlife Ecology and Conservation, University of Florida , Gainesville, Florida , United States
Amdare Nitin
Electronic publication date: 2024 Mar 29
Publication date: 2024
Volume: 12
Electronic Location ID: e16361
Received 2022 Dec 28; Accepted 2023 Oct 5
Copyright: © 2024 Gutiérrez-Ramos and Acevedo
Copyright year: 2024
Copyright holder: Gutiérrez-Ramos and Acevedo
License: This is an open access article distributed under the terms of the Creative Commons Attribution License, which permits unrestricted use, distribution, reproduction and adaptation in any medium and for any purpose provided that it is properly attributed. For attribution, the original author(s), title, publication source (PeerJ) and either DOI or URL of the article must be cited.
License URL: https://creativecommons.org/licenses/by/4.0/

Keywords: Birds, Caribbean, Heterogeneity, Haemosporidian, Puerto Rico, Virulence

Funding: FIPI (Fondo Institucional para la Investigación, translated as Institutional Funding for Investigation) at the University of Puerto Rico, Rio Piedras Campus BirdsCaribbean Hurricane Maria Relief Fund Funding was provided through a FIPI (Fondo Institucional para la Investigación, translated as Institutional Funding for Investigation) at the University of Puerto Rico, Rio Piedras Campus. A second source of funding was the BirdsCaribbean Hurricane Maria Relief Fund. The funders had no role in study design, data collection and analysis, decision to publish, or preparation of the manuscript.

==============================
Parasite transmission is a heterogenous process in host-parasite interactions. This heterogeneity is particularly apparent in vector-borne parasite transmission where the vector adds an additional level of complexity. Haemosporidian parasites, a widespread protist, cause a malaria-like disease in birds globally, but we still have much to learn about the consequences of infection to hosts’ health. In the Caribbean, where malarial parasites are endemic, studying host-parasites interactions may give us important insights about energetic trade-offs involved in malarial parasites infections in birds. In this study, we tested the consequences of Haemoproteus infection on the Bananaquit, a resident species of Puerto Rico. We also tested for potential sources of individual heterogeneity in the consequences of infection such as host age and sex. To quantify the consequences of infection to hosts’ health we compared three complementary body condition indices between infected and uninfected individuals. Our results showed that Bananaquits infected by Haemoproteus had higher body condition than uninfected individuals. This result was consistent among the three body condition indices. Still, we found no clear evidence that this effect was mediated by host age or sex. We discuss a set of non-mutually exclusive hypotheses that may explain this pattern including metabolic syndrome, immunological responses leading to host tolerance or resistance to infection, and potential changes in consumption rates. Overall, our results suggest that other mechanisms, may drive the consequences of avian malarial infection.

Introduction

Virulence, or host fitness reduction as a consequence of parasite infection, is often viewed as an unavoidable cost for parasites that reproduce at the expense of host resources (Bull, 1994; Ewald, 1994). This traditional understanding of the eco-evolutionary consequences of parasite infection comes from theoretical models that make simplifying assumptions such as homogeneous transmission (Anderson & May, 1982; Alizon et al., 2009). Still, in nature, parasite transmission is a heterogeneous process (VanderWaal & Ezenwa, 2016). Variability in parasite transmission strategies, individual traits (i.e., host immunity), and environmental factors may interact in complex ways resulting in a wide array of consequences to host health (Acevedo et al., 2019). This is particularly true in vector-borne parasite systems where infected vectors–the agents of transmission–add a layer of complexity by interacting with the host in heterogeneous environments (Lachish et al., 2011; Acevedo et al., 2019).

Haemosporidian parasites (Order Haemosporida, genera Plasmodium, Haemoproteus, and Leucocytozoon) are worldwide protists infecting birds of different families, causing a malaria-like disease (Valkiunas et al., 2014). The impacts of these parasites on birds’ host fitness are generally not well understood with empirical research showing mixed results (e.g., Møller et al., 2009; LaPointe, Atkinson & Samuel, 2012; Cornet et al., 2014; Ilgūnas et al., 2019; Videvall et al., 2020). Malarial parasites may cause detrimental effects on hosts, such as increasing mortality, and decreasing overall body condition (e.g., Atkinson et al., 1995). On the other hand, particularly where haemosporidian parasites are endemic, there might not be any negative reported effects to the hosts (e.g., Bensch et al., 2007). Multiple mechanisms have been proposed to explain this lack of negative fitness consequences including immunological strategies such as tolerance and resistance (Sorci, 2013). Within these strategies, strong negative fitness costs are avoided because the host clears the infection (resistance) or has developed an immune response that allows it to withstand infection (tolerance).

Haemosporidian parasites may have different effects depending on the age and sex of individuals, such that juveniles tend to develop a more severe infection, and even have higher mortality compared to adults (Isaksson et al., 2013). For instance, juvenile feral pigeons infected with Haemoproteus columbae are more likely to have increased infection levels and higher mortality than adults (Sol, Jovani & Torres, 2003). The naïve immune system of juveniles can be more susceptible to infection leading to higher within-host replication rates and higher parasite load (Padgett & Glaser, 2003; Calero-Riestra & García, 2016; Hammers et al., 2016). In sex-dependent studies, infected female Tawny pipits had reduced body condition compared to infected males (Calero-Riestra & García, 2016). This may be related to higher reproductive costs for females compared to males during the breeding season. Still, the sex-mediated costs of infection are not necessarily generalizable. A recent meta-analysis showed similar viability to parasitism among males and females (Hasik & Siepielski, 2022b).

In this study, we assessed the consequences of infection by avian malarial parasites on host body condition in the most abundant bird species in Puerto Rico and the Caribbean, the Bananaquit (Coereba flaveola). Specifically, we asked: (1) do Haemoproteus-infected individuals suffer from reduced body condition when compared to uninfected individuals? and (2) are related changes in body condition dependent on age or sex? We expected, following predictions from the classical theory, that infected individuals would have lower body condition and that this effect would be more pronounced in juveniles and vary by sex. If body condition decreases with infection, it would provide evidence of negative consequences of endemic malarial parasites as predicted by the theory (Alizon et al., 2009; Hasik & Siepielski, 2022b).

Materials and Methods

The Caribbean has been proposed as an ideal natural laboratory to study the ecology and evolution of vector-borne parasite-host interactions (Ricklefs et al., 2017). In the Caribbean, malarial parasites are endemic, and host species diversity is low, but many species are generalists occupying a wide variety of habitats (Acevedo & Restrepo, 2008). These factors create a unique set of conditions for host-parasite co-evolution. While multiple studies describe the biogeographic patterns of malarial parasites of Caribbean bird hosts (e.g., Fallon et al., 2004; Ricklefs et al., 2017), our understanding of the potential health consequences of malaria infection to Caribbean bird hosts is limited.

We conducted the study from June 2018 to January 2019 in 13 urban forest patches (each site was visited 1–4 times, Table SI) in the metropolitan area of Puerto Rico, an urbanized area that comprises 10% of the island (Fig. S1; Table S1) (Martinuzzi, Gould & González, 2007). In Puerto Rico, Bananaquits breed throughout the year with increased reproductive activity between February and June (Wunderle, 1982). We chose Bananaquits (Coereba flaveola) as our study species because it is the most abundant species in urban forests in the Caribbean and previous studies showed that Plasmodium and Haemoproteus parasites commonly infect this species (Wolff et al., 2018, Antonides et al., 2019). Note that, while we use the basal genus Haemoproteus to describe the parasites in this study, it is likely that these lineages belong to the Parahaemoproteus subgenus (Martinsen, Perkins & Schall, 2008). All bird handling procedures were conducted with approval of the Institutional Animal Care and Use Committee (IACUC) of the University of Puerto Rico protocol number 3011-02-05-2018, the USGS Federal Bird Banding Permit number 21669, and the Department of Natural and Environmental Resources of Puerto Rico permit number 2018-IC-066. All individuals in this study were captured using 2.5 m × 6 m and 2.5 m × 12 m mist nets. We used four to eight mist nets per sampling period. Nets were open by sunrise and closed between 8:30–10:00 AM depending on sunlight, weather, or presence of raptors. Bananaquits are difficult to sex and/or age due to their monomorphic plumage. Upon capture, we aged and sexed individuals using standard procedures such as visual inspection of cloacal protuberance, brood patch, and skull pneumatization (Ralph et al., 1993). Individuals with a defined brood patch were classified as female and individuals with a prominent cloacal protuberance were classified as males (Ralph et al., 1993). We measured wing length and tarsus length as parameters for body condition index to the nearest 0.1 mm (Wunderle, 1994). Also, we measured bird body mass as another parameter for body condition index to the nearest 0.1 g to estimate residual body condition indices. After taking measurements and extracting a small blood sample, we released all the individuals back to their habitat.

To diagnose infection status, we extracted 10–30 µl of blood from the brachial vein, which we collected on filter paper and stored at 20 °C. We extracted DNA from the blood samples using DNeasy Blood & Tissue Kit (Qiagen Hilden, Germany) and performed a nested PCR to detect parasite presence in the avian host species. We used two sets of primers, the Haem primers and the MalUniv primers (S. Perkins, 2019, personal communication) to increase the probability of detection (Fallon et al., 2003; Hellgren, Waldenström & Bensch, 2004). We used the diagnostic standard protocol established by Hellgren, Waldenström & Bensch (2004). For Mal Univ, we used 10 µl of TopTaq Polymerase, 1 µl of MalUnivF primer, 1 µl of MalUnivR primer, 2 µl of coral load, 5 µl of nuclease-free water, and 1 µl of DNA template per reaction (SI Article). Positive samples were detected when a band appeared in the electrophoresis gel at 500 bp and negative when no band was found (Fig. 1A). We purified positive samples using the Qiagen PCR purification kit. Positive samples were sequenced using the Sanger Sequencing Service at the Sequencing and Genotyping Facility (University of Puerto Rico, Rio Piedras). We analyzed the sequence using Mega X software (Kumar et al., 2018) and then used the BLAST database (https://blast.ncbi.nlm.nih.gov/Blast.cgi) to determine the parasite genus. Then, we extracted the results that had 97% or more identification accuracy to identify the avian malaria parasite genus. All the infected were classified as Haemoproteus, Plasmodium, or as unclassified positive when we could not determine based on the base pairs in the sequence. Given the limited number of Plasmodium-infected individuals (n = 3), we restricted our analysis to Haemoproteus infections.

Figure 1 Diagnostic techniques of Haemoproteus parasite infection in the Bananaquit.

(A) Sample 095 is positive for haemosporidian infection with a band at approximately 500 bp. Bands at 50 bp are primer dimers, a by-product of the PCR. (B) Image shows a Giemsa-stained slide showing an infected erythrocyte of sample 095. Red arrow indicates an erythrocyte infected by an haemosporidian. Although we show a Giemsa-stained slide, all analyses were based on molecular diagnostics.

Statistical analysis

To test if body condition decreased with infection by Haemoproteus, we quantified three types of body condition indices, two residual indices of body mass and a body size measurement, and a principal components analysis (PCA). All the statistical analyses were conducted using the infection status results from molecular diagnostics. We used a residual body condition index by analyzing residuals of a linear relationship between natural log body mass predicted by natural log wing length, and the linear relationship between natural log body mass predicted by natural log tarsus length (SI Article). Residual indices describe body condition as a function of the relationship between body length and mass (Peig & Green, 2010). The residual is the difference between the observed and predicted values (Larsen & McCleary, 1972). Individuals with residuals above zero are considered to have higher body condition than average and individuals with residual values below zero have poorer body condition than average. These body condition indices are commonly applied in similar studies to assess the consequences of malarial parasites on birds’ health (e.g., Brock et al., 2013; Marzal et al., 2015).

We also tested for a decrease in body condition by avian malaria infection using a principal component analysis (PCA) which synthesizes multiple correlated variables such as mass, wing length, and tarsus length into correlated axes (Peig & Green, 2010). In the PCA, the data were scaled and centered. We used the PC1 loadings as an index of body condition because PC1 explains most of the variance compared to other axes (see results section). This type of analysis is commonly applied to assess the potential negative effects of infection on hosts’ health (e.g., Hatchwell et al., 2001). Still, some studies suggest caution when applying body condition indices because these are highly dependent on the body measurements used to calculate the index (Sánchez et al., 2018). To address this issue, we applied these three complementary types of body condition indices. If the results of the indices are consistent it would suggest that the result is robust. We also tested for the distributional assumption of normality of these parametric models using a Shapiro-Wilk test. To determine if body condition changed with infection status and age or sex, we used linear models with infection status and an interaction effect with age (hatch-year or juveniles-HY or after hatch-year or adults-AHY), or sex (male or female) as a predictor for body condition. Note that we conducted different models for age and sex and only on individuals with known sex or age, and infection status. We tested the need to add the mist-netting site as a random effect using a likelihood ratio test. We conducted the statistical analysis using R statistical software v4.3.0 (R Core Team, 2023). We used the following packages for data organization, analyses, and visualization: “ggplot2” (Wickham, 2016), “dplyr” (Wickham et al., 2023), “lme4” (Bates et al., 2015), “lmerTest” (Kuznetsova, Brockhoff & Christensen, 2017), “Matrix” (Bates, Maechler & Jagan, 2023), “tidyverse” (Wickham et al., 2019), “devtools” (Wickham et al., 2022), “ggbiplot” (Vu, 2011), “sjPlot” (Lüdecke, 2023), “sjmisc” (Lüdecke, 2018), “sjlabelled” (Lüdecke, 2022), “snakecase” (Grosser, 2019), “RColorBrewer” (Neuwirth, 2022), “RLRsim” (Scheipl, Greven & Küchenhoff, 2008), “ggpubr” (Kassambara, 2023), “olsrr” (Hebbali, 2020) and “effects” (Fox & Weisberg, 2018).

Results

Body condition and infection status

We captured a total of 79 Bananaquits and collected blood samples from 66 individuals. Thirteen individuals were not included in the analyses either because they escaped before the processing was completed, available blood after puncturing the brachial vein was insufficient, or because the blood coagulated in the capillary tube. Out of these, 47 were classified as adults (AHY), 13 were classified as juveniles (HY) and two were unidentified (U) (Table S2). From the total captured, we were able to classify 19 as male and 10 as females. A total of 18 individuals were detected through molecular diagnostics as infected by haemosporidian parasites for an overall prevalence of 27% from the sampled population. Most infections corresponded to the genus Haemoproteus (n = 15) followed by parasites from the genus Plasmodium (n = 3) and three unclassified positives.

As expected, the linear regression model of log(weight) as a function of log(wing length) showed a clear positive relationship (t = 6.20, p < 0.001, R2 = 0.39; Table S3A). Similarly, the linear regression model of log(weight) as a function of log(tarsus length) also showed a strong positive relationship (t = 2.98, p < 0.01, R2 = 0.13; Table S3B). Tarsus and wing length were moderately correlated (r = 0.32).

On average, the body condition of infected Bananaquits was higher than uninfected ones (0.05 ± 0.02 SE) when comparing body weight relative to wing length (t = 2.16, p = 0.04; Fig. 2A; Table S4A). Similarly, the body condition of infected Bananaquits was higher than uninfected ones (0.07 ± 0.03 SE) when comparing body weight relative to tarsus length (t = 2.59, p = 0.01; Fig. 2B; Table S4B). The linear mixed-effects model for wing length showed a singular fit and the likelihood ratio test comparing the tarsus-length model with and without a random effect for site show no clear evidence for the need for this random effect (LRT = 0.20, p = 0.21). Therefore, we made the inferences above using fixed effects models.

Figure 2 Comparison between body condition of uninfected and infected Bananaquits by Haemoproteus parasites using two residual body condition indexes and a PCA body condition index.

Comparison of body condition of uninfected (n = 47) and infected by Haemoproteus (n = 15) Bananaquits using a (A) wing and weight residuals index, (B) tarsus and weight residuals index and (C) PCA body condition index. The horizontal dash line at zero represents the average body condition. Individuals above the line have higher body mass than average, while individuals below the line have lower body mass than average. Jittered dots indicate individuals included in the analysis and their classification as non-infected and infected individuals. The lines indicate the 95% confidence intervals, dot indicates the point estimate of the model of the non-infected individuals and triangle indicates the point estimate of the model of the infected individuals.

Although the distribution of wing length deviated slightly from a normal distribution, we kept it in the PCA model after inspecting the histogram and quantile-quantile plot that showed just small deviations from normality (Figs. S2–S4). The first axis, PC1, explained 62.7% of the variance, while PC2 explained 24.6%. Similar to the residual body condition index, the model predicting PCA body condition index as a function of infection status showed that infected Bananaquits had higher body condition (b1 = 0.96 ± 0.40 SE) than uninfected individuals (t = 2.43, p = 0.02; Fig. 2C, Table S4C).

Individual heterogeneity in body condition by infection status

We did not find evidence that the effect of infection on the Bananaquit body condition varied by age (N = 60) or sex (N = 29). Age did not significantly contribute to variation in the body condition index either using weight relative to wing (interaction: t = −1.01, p = 0.32; Table S5A) or tarsus length (interaction: t = −0.11, p = 0.91; Table S5B), or using the PCA body condition index (interaction: t = −0.31, p = 0.76; Table S5C; Fig. S5). Similarly, sex did not significantly contribute to variation in the body condition index using weight relative to wing (interaction: t = −0.61, p = 0.55; Table S6A) or tarsus length (interaction: t = −0.02, p = 0.98; Table S6B), or using the PCA body condition index either (interaction: t = 0.51, p = 0.62; Table S6C) (Fig. S6).

Discussion

Many studies have described the negative consequences of malarial infection to naïve bird populations (LaPointe, Atkinson & Samuel, 2012). Still, we know little about the consequences of infection in regions where the parasite is endemic and infections chronic. While a decrease in host survival, fecundity or other sub-lethal measures are an expected outcome of many parasitic infections (Hasik & Siepielski, 2022b), our results showed that infected individuals had higher body condition than uninfected ones. We found no clear evidence that these effects varied with sex or age. Therefore, our results suggest that there may be alternative underlying mechanisms that do not necessarily result in negative consequences for body condition. Three non-mutually exclusive hypotheses may explain this result: (1) metabolic syndrome that predicts higher fat storage in infected individuals, (2) host tolerance or resistance to infection, and (3) changes in foraging behavior.

Parasite infection can trigger immunological responses that often lead to inflammatory reactions, a decrease in muscle performance, and increased levels of carbohydrates in the blood, which is commonly known as the metabolic syndrome (Schilder & Marden, 2006). Excess carbohydrates and lipids can explain why some individuals have a higher body condition (i.e., larger mass relative to the average). In birds, accumulated lipids provide extra energy storage to survive long-distance travel (Guglielmo, 2018) and in some species like the Blue Petrel is associated with improved reproductive success (Chastel, Weimerskirch & Jouventin, 1995). Still, during our sampling period, our study species showed little to no body fat accumulation (N. Gutierrez, 2019, personal observations) a parameter that is highly variable depending on the Bananaquit habitat (Douglas, Winkel & Sherry, 2013; Bergstrom et al., 2019). Therefore, while metabolic syndrome may explain fat accumulation in other species, there is no strong evidence supporting this hypothesis in our host-parasite system.

There are examples of bird diseases such as malaria and Mycoplasma in which populations of naïve hosts initially suffer high mortality due to parasite infection followed by a population-level decrease in these negative consequences due to resistance or tolerance traits (Sorci, 2013). Tolerant individuals do not reduce or clear the infection but have mechanisms to reduce their negative effects on their survival, reproduction or other sub-lethal effects (Medzhitov, Schneider & Soares, 2012). Hence, tolerant individuals suffer small to no parasite-induced mortality or changes in body condition. For instance, tree swallows and eastern bluebirds’ nestlings show no decrease in survival when infected by the parasitic flies (Protocalliphora sialia) (Grab et al., 2019). In rodents, it has been shown that infection by macroparasites leads to increased body condition (Jackson et al., 2014). Alternatively, resistant individuals reduce or clear parasite infection by activation of innate and adaptive immunological responses (Medzhitov, Schneider & Soares, 2012). Contrary to tolerance, resistance can be costly to host fitness because it often results in tissue damage through the immunological response activation to eliminate the pathogen (Medzhitov, Schneider & Soares, 2012). For instance, a study of the Seychelles warbler found that individuals’ infection status was related to reactive oxygen metabolites (ROMs; van de Crommenacker et al., 2012). During the breeding stage, ROMs were significantly higher in infected individuals compared to non-infected individuals, which may indicate an immunological activation and/or the metabolic residual of the parasite infecting the individuals. Higher body condition of infected individuals compared to uninfected individuals may suggest a tolerance mechanism on the host because body condition parameters show no negative effects on host physiology (Atkinson et al., 2013). It is likely that, in the Caribbean malarial parasites and the Bananaquit have co-evolved, and the parasite may have adapted to exploit resistant and/or tolerant individuals that ultimately lead to higher parasite transmission rates (Metcalf et al., 2012). Nevertheless, the mechanisms by which tolerance and resistance evolve are still not fully known. There is some evidence of genes related to immune function responding to selection pressures from vector-borne parasites (e.g., Bonneaud et al., 2012). One related hypothesis is that we are more likely to trap chronically infected individuals because acutely infected individuals may suffer from lower mobility (Mukhin et al., 2016) and, thus, may be less likely to be trapped in our mist nets. Thus, if this parasite causes severe negative consequences to host health these individuals would die quickly or are less mobile and hence, they will have lower capture rates and be underrepresented in our samples. While our data does not allow us to differentiate between tolerance, resistance, or trapping bias, these remain key alternative hypotheses to test in future studies. For instance, previous studies have experimentally infected individuals to track their immune responses and other physiological factors through peak infection and beyond (Adelman et al., 2013).

As a response to offsetting the cost of infection, host species may change behavior by increasing their foraging activity which can ultimately result advantageous to the host and the parasite (Weinersmith & Earley, 2016). While a recent meta-analysis showed that parasite-infected hosts consume on average 25% less food than uninfected individuals, the study showed great variability among taxa with multiple examples of the opposite pattern (Mrugała, Wolinska & Jeschke, 2023). For instance, parasite-infected rusty crayfish consume more macrophytes and macroinvertebrates than uninfected ones likely due to increased feeding behavior boldness induced by infection (Reisinger & Lodge, 2016). Also, hosts with access to more or higher quality food resources would have on average better body condition and thus experience higher parasitism rates because they are optimal hosts for the parasite. Indeed, Hasik & Siepielski (2022a) found that hosts with access to more prey were more heavily parasitized, though they did not relate this increased parasitism to the quality of the host for the parasite. The relationship between cost of infection and feeding behavior is likely mediated by food availability. Our study species, the Bananaquit, is a generalist and it is highly adapted to exploit a wide variety of food resources. Therefore, the availability of a wide variety of food resources combined with a potential increase in foraging activity as a compensatory behavioral response to infection (Ots, Murumägi & Hõrak, 1998; Sorci, 2013; Toscano, Newsome & Griffen, 2014) may also explain an increase in body condition. Natural disturbance can also have a mediating role in the effects of parasite infection on hosts’ health (Sousa, 1984). In September 2017, Puerto Rico suffered the impact of a strong category four hurricane that devastated a large portion of the island causing high mortality in flora and fauna, including birds (Wunderle, 2017). This high mortality event could have served as a strong selection event favoring individuals with traits related to enhanced physiological performance or immunity (e.g., Donihue et al., 2018). Therefore, if there existed a pool of weaker individuals that would have shown strong negative effects of body condition due to malarial infection, these may be underrepresented in the host population in the aftermath of the hurricane. This assumes that traits related to survival to large-scale disturbances are also related to immune response to parasites which may not necessarily be the case but still is a hypothesis that remains to be tested.

Previous studies have shown variable effects on individuals after natural disturbances. For instance, a study of Cerulean warblers’ responses to simulated natural disturbances in the Appalachian Mountains showed that males in areas of less disturbance had better body condition compared to males in areas of heavy disturbance (Boves et al., 2013). In contrast, amphibians showed a reduced risk of Bd (Batrachochytrium dendrobatidis) infection in areas with higher canopy openings resulting from Cyclone Yasi compared to undamaged areas (Roznik et al., 2015). Therefore, hurricane disturbance may have been a factor that mediated the overall susceptibility of the bird host population.

Lastly, we found no clear statistical evidence of differences in infection status due to sex or age. There are strong theoretical arguments to suggest that sex and age of the infection are key determinants of quantifiable virulence traits of the host (Day, 2003; Frank & Schmid-Hempel, 2008; Lively, 2010). There is also empirical support for this theoretical idea in many host-parasite systems (e.g. Sorci & Faivre, 2022; Izhar & Ben-Ami, 2015; De Roode, Gold & Altizer, 2006). There are two potential explanations for the lack of evidence in our study. Our sample size was limited when dividing the data among sex or age. Therefore, if the effect size of the effect of infection by these covariates in this system is small, the analyses may not have enough power to detect them. Alternatively, there may not necessarily be a strong difference between these traits. For instance, a recent meta-analysis on parasitism and host fitness variation also shows no clear difference by sex (Hasik & Siepielski, 2022b) but, they found high variability and some studies in avian malaria still have found differences in infection rates by sex (e.g., Calero-Riestra & García, 2016). Thus, potential heterogeneities in virulence due to age or sex are still a plausible hypothesis worth further consideration.

Conclusions

Our study provides insights into the consequences of malarial infections to the most common bird in the Caribbean showing that infected individuals had higher body condition compared to uninfected individuals. Multiple hypotheses can explain the pattern in our system including tolerance in infected individuals. The tolerance hypothesis is an interesting explanation for the observed patterns in our study that can be further tested by conducting controlled infection experiments.

Supplemental Information

Supplemental Information 1 Data used for analysis of infection status and body condition of Bananaquits.

Supplemental Information 2 R code for analysis of infection status and body condition of Bananaquits.

Supplemental Information 3 Sequences.

Supplemental Information 4 Supplemental Information

We thank T. Mitchell Aide and J.M. Wunderle Jr. for their feedback and guidance on this research project. We also thank our dedicated field technicians A. Medina, A. Rodríguez, and E. Hernández and J. Rodríguez for providing the molecular laboratory space. Finally, We thank A. Miró-Herrans and A.G. Rivera-Colon for providing guidance with the analysis of the sequences.

Additional Information and Declarations

Competing Interests

Author Contributions

Animal Ethics

Ethics

Field Study Permissions

DNA Deposition

Data Availability

The authors declare that they have no competing interests.

Nicole A. Gutiérrez-Ramos conceived and designed the experiments, performed the experiments, analyzed the data, prepared figures and/or tables, authored or reviewed drafts of the article, and approved the final draft.

Miguel A. Acevedo conceived and designed the experiments, analyzed the data, prepared figures and/or tables, authored or reviewed drafts of the article, funding, and approved the final draft.

The following information was supplied relating to ethical approvals (i.e., approving body and any reference numbers):

The Institutional Animal Care and Use Committee (IACUC) of the University of Puerto Rico provided full approval for this research (protocol number 3011-02-05-2018).

The following information was supplied relating to ethical approvals (i.e., approving body and any reference numbers):

The USGS Federal Bird Banding Lab provided the permit for capture and release of birds (Permit number 21669).

The following information was supplied relating to field study approvals (i.e., approving body and any reference numbers):

Field experiments were approved by the Department of Natural and Environmental Resources of Puerto Rico (permit number 2018-IC-066).

The following information was supplied regarding the deposition of DNA sequences:

The sequences are available at GenBank: OQ127425 to OQ127439.

The following information was supplied regarding data availability:

The code and raw data are available in the Supplemental Files.

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
