# Peer review of "Higher body condition with infection by Haemoproteus parasites in Bananaquits (Coereba flaveola)"

_PeerJ, doi:10.7717/peerj.16361_

## Round 0.1 · original submission · Major Revisions

Your PeerJ manuscript entitled "Better body condition with infection by Haemoproteus parasites in the Bananaquit (Coereba flaveola), a resident species of Puerto Rico" has been reviewed.

The reviewers were of the opinion that the manuscript contains important information of interest to other investigators. However, they also identified several concerns that require further attention, as indicated in the reviews appended below.

I noticed that the referees suggested some revisions/additions to the introduction part, and/or I suspect that the suggested text revisions might also lead to revisions of your results, methodology, and discussion also. The methods regarding the infection status need to be more explained. Please consider those suggestions and make appropriate changes.

Reviewer 1 ·

Basic reporting

Overall impression
In this study the authors attempted to understand how infection with malaria affects the health of its avian host, the Bananaquit. They do this by sampling wild birds for infection with the parasite, and then relating their infection status to host body condition.
The strengths of this study are the use of wild, semi-urban populations. Using animals from the field allows the authors to understand how these patterns of infection occur in nature. A further strength is the authors using multiple indices of body condition to ensure that their results are robust.
The only weakness that I can find with the study is the limited sample size, but this is an understandable limitation and does not take away from the relevance and importance of the results the authors found.
Below I outline major and minor comments that I hope will be useful to the authors. I think that this manuscript could be a good fit for PeerJ after addressing these comments.

Major Comments
1. My only major comment is that a discussion of sex-differences (or the lack thereof) may improve the discussion. A recent meta-analysis found that males and females suffer similar parasite-mediated decreases in fitness (Hasik & Siepielski, 2022 Biology Letters). A paragraph in the discussion related to this point may improve the impact of this paper by making it more comparable and applicable to the results from other host-parasite systems.
Hasik, Adam Z., and Adam M. Siepielski. "Parasitism shapes selection by drastically reducing host fitness and increasing host fitness variation." Biology Letters 18, no. 11 (2022): 20220323.


Minor Comments

1. Lines 47-48: Surely there has been more work dedicated to understanding how avian malaria affect bird host fitness in the ten years since this paper was published? If not, please revise to include more recent papers, or cite the appropriate (and more recent) papers showing if/how avian malaria affects bird hosts, even if those results are mixed. The authors could explain that evidence is mixed using the paper they already cite, but then include further papers showing that the results are still inconclusive. Parasites tend to severely reduce host fitness (Hasik & Siepielski 2022, Biology Letters), so if this doesn’t apply to avian malaria please provide papers/justification for why it does not.
2. Lines 62-63: While the reduced body condition in females that was cited may be a system-specific response to infection, the aforementioned meta-analysis (Hasik & Siepielski 2022, Biology Letters) showed that the parasite-mediated reductions in reproductive fitness (specifically viability and fecundity) is similar among males and females.
3. Lines 102-103: Did you capture these birds during the breeding season? I assume you did, but it is unclear as written.
4. Line 155: please explain what HY and AHY represent on this line, and not later in the results on line 160, to improve the flow of the paper.
5. Lines 168-172: I don’t think the authors need to mention that the residuals are normally-distributed.
6. Lines 176-179: same comment. There’s no need to justify the use of parametric models, and the authors could save space and make the paper more concise by removing these statements.
7. Line 184: is this the difference between the infected and uninfected individuals, as above? Or is this just the mean and SE of the infected birds? Please clarify, and if it’s the latter then please add the mean and SE for the uninfected birds as well.
8. Line 196-198: This statement on assumptions should also be removed.
9. Lines 209-210: You already abbreviated PCA, you don’t need to explain what it is again
10. Lines 236-238: I really like this bit of the discussion! Great way to tie this to other systems while also justifying why it is not relevant in your system.
11. Lines 264-266: Could this not instead be a sign of better body condition leading to better immune defenses, thus the ones in better condition survive infection, while you don't detect the infected birds that are in bad condition because they die?
12. Lines 300-301: Studies in aquatic host-parasite systems have also found that hosts provided access to more food (which should result in better body condition) had higher parasitism on average (Hasik & Siepielski, 2022 Freshwater Biology), when the authors expected the opposite pattern.

Hasik, Adam Z., and Adam M. Siepielski. "A role for the local environment in driving species‐specific parasitism in a multi‐host parasite system." Freshwater Biology 67, no. 9 (2022): 1571-1583.

Experimental design

No comment.

Validity of the findings

No comment.

Additional comments

No comment.

Reviewer 2 ·

Basic reporting

The study entitled “Better body condition with infection by Haemoproteus parasites in the Bananaquit (Coereba flaveola), a resident species of Puerto Rico” analyzed the relationship between infection with body condition, age, and sex. They found that infected birds had a higher body condition and found that sex and age did not relate to the infection status. I believe that it suits the PeerJ journal, but I have three main comments:

1) I believe the introduction needs to focus on the trade-off between infection and host life-history traits and immunity. The authors need to bring information about how individuals deal with infection by directing more resources to immunity and less to other physiological pathways. However, some individuals may be able to invest in immunity without compromising their body condition. There are quite a lot of studies stating this, so I believe that it would not be difficult to make this point
2) The methods regarding the infection status need to be more explained:
a) What type of PCR was performed?
b) Did you measure the slides? I saw one picture about it so that makes me wonder about the slides
c) You removed the Plasmodium-infected birds, you would need to state the reasons, and state the sample size per model since it would be different from the initial presented value.
d) You can state the stats about body condition (individual measurements), and PCA in the methods. The results should focus on the model results (GLMM models).
3) The discussion needs to be rephrased. I made several comments to direct another path. Go back to point 1 above and try to state with more examples the trade-off between immunity, physiology, and life-history traits. There are more studies you can use, to talk about the vulnerability to parasitism.

Title:
I would remove the word “better” and state “higher”. Also, I would remove “a resident species of Puerto Rico”. Title suggestion: Infected bananaquits (Coereba flaveola) had higher body conditions in Puerto Rico

Abstract
Line 13: I would remove “the reduction in host fitness due to parasite reproduction at the expense of the host resources” and “classic”.

Line 16-24: I believe that you mentioned a lot about the introduction here, but the methods, results, and discussion were not well balanced.

Line 26: Surprisingly to whom? I would remove this word here.

Line 29-31: This needs to be rephrased. I made several comments in the discussion part, and after those are taken care of, you can add more information here.

Introduction
Line 35: I wouldn’t call this virulence if that is what you are trying to convey here. I would simply remove “or host fitness reduction due to parasite infection”

Line 39: What do these expressions mean (horizontal contact transmission and homogeneous mixing)? If you are bringing up those terms early in the introduction, I believe it would be great to define them.

Line 40: What do you mean by inherently heterogeneous process? This expression was a bit difficult to follow. If you are trying to say that transmission is not homogeneous, just state that with a couple of examples.

Line 41: Be specific with the expression “transmission strategies”. Just add a parasite to it, so that it becomes: a parasite transmission strategy.

Line 42: Virulence also depends on certain aspects of the parasite as well as the immunity of the host. I would remove this part, and just say that parasite transmission strategies, (host or parasites?) life-history traits, and climate may affect parasite-host interactions.

Line 44: What do you mean by heterogeneous ways? Be more specific. Maybe give an example here to close the paragraph.

Line 48: Only one study is cited here to say that it is not well understood. I would be more careful here and remove this sentence.

Line 50: What kind of selection pressure are you talking about? I would get the specific terms that they used in their manuscript here.

Line 55: Remove “suffer” and place “are more likely infected and with higher mortality than adults”.

Line 56: In which terms are you referring to stress here? Be more specific. Parasitized juveniles with higher corticosterone are more likely ….  something like this.

Line 60: I believe you made a drastic change to your body condition. Introduce the term first and say how it relates to parasitism. Then, state the examples.

Line 62: It lacks a “to” after related.

Line 68: Why does it create a good condition for that? Just state your reasons.

Line 71: place “ed” after assessment.
Line 73: Compound adjectives: just write: Haemoproteus-infected individuals

Line 74: I would be careful with the term induced. If you are looking at a long-term study, and you are also evaluating how the body condition changes with regards to infection, then that would be ok.

Line 75: Rephrase this sentence so that it reads: “We studied the relationship between infection by haemosporidian parasites and body condition in bananaquits (Coereba flaveola)”.

Line 79: It lacks some citations here if you are referring to classical theoretical studies.

Line 79: I understand that you tried to cover as much ground as possible here, but it got confusing. Remove the last two sentences. Be aware that the relationship between body condition and parasitism may be opposite in a way that two different questions are possible:
a) Are ill-equipped birds, with lower body conditions, more vulnerable to diseases? It means that birds are already in a poor condition, so they are more vulnerable not only to diseases but other sorts of predation.
b) Do infected individuals trade investing in immunity in body reserves to fight off the disease? It means that birds may have their body condition reduced because they would direct more resources to immunity and less to plumage coloration, body condition, and even reproduction.
Please note that both scenarios are possible. If you pick one, state that clearly with specific examples since we have enough studies directed by both cases.

Experimental design

Methods
Line 88-91: I would remove this part. I would make this part more concise, to something like We studied the Bananaquit because of its high abundance in urban locations in Puerto Rico, and due to its high likelihood of infection, as demonstrated by other studies (CITE A, B, C).

Line 102: Could you age and sex all individuals along the season? If yes, state here, if no state how many birds you failed in doing that. You state the difficulty right after this sentence. It means that you only captured birds during the breeding season. If so, state here.

Line 105: How did you estimate body condition? Scale-mass index? I know you specified below, but just quickly state it here.

Line 107: What was the good condition for release? Do you have a protocol for that? I would simply remove this information.

Line 109: How did you bleed the birds? Needles and capillary? Brachial vein, tarsus?

Line 112: What kind of PCR did you do? Nested? More information is needed.

Line 125: What do you mean by limited numbers? Did you remove those infections? How many birds did you remove? I believe that you need to give more information here.

Line 131-135: Just use the equation here. It is much easier for the reader to follow the clear equation. Also, you analyzed the residuals of a linear REGRESSION between mass and body length. So, you need to include the stats here (r2, p-value) of that regression. Also, how many times did you use the wings? Did you have any repeatability measurements?

Line 141: Were these variables (body condition) correlated? I did not see the stats here.

Line 146: How much is the accumulated variation of PC1 and PC2? Also, PC1 is more related to which variables? All? You can add this information in the supplementary material.

Line 149: What do you mean by variables included? State here.

Line 153-156: Where did you do these analyses? R? Which packages did you use? Also, how did you select the best model? Did you compare this full model with a null model, or did you simply do a backward or forward model selection here? What did you base our decision on? AIC, p-value, BIC?

Validity of the findings

Results
Line 159: Why weren’t you able to get blood samples from all birds? 1-2 I would assume that it would be ok, but 16 birds… I would why?

Line 169: I would simply place this information about the body condition in the methods section. Here, just state the details about the models.

Line 176-179? Again, this may be placed in the methods section. I would only keep the information about the LM or GLM.

Line 181: Are you sharing these histograms as well? If you say that you visually inspected things, that would be great to show the graphs.

Line 181-187? I would again leave this to the methods.

Line 188-198: I would change this paragraph to something like: We did not find evidence that age (STATS), sex (STATS), tarsus length (STATS), body condition (STATS)… Also, the title says infection, and I do not see any results for infection here.

Discussion
Line 204: I would write “higher” body condition instead of better.

Line 208-2012: Methods, I would remove this paragraph.

Line 215: remove the larger

Lien 219: State which hypothesis the paragraph is going to be talking about.

Line 219 – 238: So, if you are saying that there is not enough size variability in the study system, I would not discuss the body size as a potential factor here. I would suggest removing this portion of the discussion.

Lien 242: What do you mean by chemical changes? Do you have anything more specific here?

Line 239-255: I would deal with this paragraph kindlier. This is an important part of the discussion. I would remove that it does not help and get more studies analyzing the fact that a higher body condition can enhance individuals' for dealing with parasitism by haemosporidian parasites. What happens before getting infected? This is the breeding season, birds are paring, finding territories, and some are singing…. So, try to discuss this fact, since this would give the reader a better perspective of the moment that the bird is going through.

Line 274: I would say that the vectors may have not survived, and therefore, some of the haemosporidian parasites.

Line 274-279: I would be more careful in this paragraph. There are two important pieces of information here:
a) Hurricane: this natural disaster may affect the likelihood of infections. Bring more studies that relate natural diseases to infection rates. This might help us understand this pattern here.
b) Since you did not measure resistance and tolerance, I would be more careful here. State that better-equipped birds may be able to mount the immune system against parasites without compromising their body condition. Therefore, these individuals can have a high body condition and be infected at the same time. Get examples that talk about this, relating to plumage coloration, reproduction…

Line 280-286: Even though this is a great study, it does not help much here. Also, I thought it was a bit lost in the middle of the paragraphs here. I would remove it.

Line 287-301: I would merge this paragraph with the one in 274-279, to make it a stronger argument.

Conclusions
I missed a bit about the discussion here. After remodeling the discussion, I would like to see a reason why you found infected birds with higher body conditions.

Figures
1 - Did you do slides? Do you have data on parasitemia? If yes, that would be great to make an input in this paper.

2 – What are the colors saying here? I would remove the colors and keep the definitions on the x-axis here.

---

## Round 0.2 · Major Revisions

Dear Author,

Thank you so much for submitting the revised version of the manuscript. Both reviewers have acknowledged the importance of your study and its potential significance in the field. One of the reviewers expressed mostly positive feedback on the revisions you made and suggested minor revisions, considering the manuscript's importance in the field. However, the other reviewer has requested major revisions, highlighting the following concerns:

1. Use of generalized wordings: The reviewer has noted that certain sections of the manuscript contain generalized language. Providing more specific and precise descriptions is recommended to enhance clarity and precision in conveying your research findings.

2. Detailed information about the statistical methodology used: The reviewer has requested additional information regarding the statistical methodology employed in your study. Please consider providing a more comprehensive explanation of the statistical methods used.

3. Clarity about age and sex-related aspects.

Please address the concerns raised by both reviewers in your revised manuscript. Once again, we appreciate your efforts in revising the manuscript and look forward to receiving the updated version.

Reviewer 1 ·

Basic reporting

Overall impression
In this study the authors attempted to understand how infection with malaria affects the health of its avian host, the Bananaquit. They do this by sampling wild birds for infection with the parasite, and then relating their infection status to host body condition.
Having already reviewed a previous version of this paper, I only have minor comments on this revision. I feel that authors have suitably address all of my previous comments.
Below I outline one major and a few minor comments that I hope will be useful to the authors. I think that this manuscript will be a good fit for PeerJ after addressing these comments.

Major Comments
1. Lines 272-274: While the authors of Hasik & Siepielski 2022 Freshwater Biology did find that parasitism increased with prey/resource density, they did not connect this to host quality. Indeed, they actually made the point that the increased parasitism may be because the parasite, not the host, benefitted from the increased resources. I appreciate the point that the authors are trying to make with this statement in the Discussion, but this needs to be reworded so that is accurately reflects the paper they are citing. The authors can even keep this statement that they have, but I suggest editing it to reflect the findings of Hasik & Siepielski 2022 Freshwater Biology. That is, say something like “Also, hosts with access to more or higher quality food resources would have on average better body condition and thus experience higher parasitism rates because they are optimal hosts for the parasite. Indeed, Hasik & Siepielski (2022) found that hosts with access to more prey were more heavily parasitized, though they did not relate this increased parasitism to the quality of the host for the parasite.”


Minor Comments

1. Line 50: I’d change “individual traits including host immunity” to “individual traits (i.e., host immunity)” to improve the flow of this sentence.
2. Line 70: I would change this to “usually quantified through” to avoid repeating “often” in the same sentence.
3. Lines 90-91: I’d change this to “reduced body condition when compared to uninfected individuals?”
4. Line 95: Change to “are at play” to avoid repeating “may” in the same sentence.
5. Line 195: Change to “while PC2 explained 24.4%”.
6. Line 214: Change to “that these effects varied with sex or age”.
7. Lines 317-318: Change to “by conducting controlled infection experiments”.

Experimental design

No comment

Validity of the findings

No comment

Additional comments

No comment

Reviewer 2 ·

Basic reporting

General comments
Gutiérrez-Ramos and Acevedo with their study entitled “Higher body condition with infection by Haemoproteus parasites in the Bananaquit (Coereba flaveola)” analyzed the effects of age, sex, and body condition on the parasite occurrence. I feel like this is an important study performed in Puerto Rico and the audience from PeerJ may benefit by reading this study. However, I have six main comments:
1) Be careful with the wording. I feel like there is some generalizations using the word fitness, virulence, tolerance, and resistance. I indicated a couple of times where you can be more specific and avoid misunderstandings. Also, if you measured the presence or absence of parasites, I would use the expression parasite occurrence overall, and not virulence.
2) There are current studies treating Parahaemoproteus as a separate genus. You might want to do the same, since I hardy believe that you had Haemoproteus Haemoproteus, but instead Haemoproteus Parahaemoproteus. There are a couple of recent studies doing this, so I suggest you do the same.
3) How did you sex individuals? Only by the brood patch? If you said that they mostly breed from June to January, and you captured the individuals from January to June, how reliable would the brood patch be to sex individuals?
4) I think there are more information needed in the statistical analysis part: did you test for an interaction between sex and age? Since you found unknown sex, age, and presence of parasites, did you use these categories in the models? If yes, I would be highly inclined to remove those, since these might not have any biological meaning. Also, how many individuals did you use in every analysis. Cite the sample size in parenthesis every time, please. In addition, apparently you ran 6 different models, so indicate this number in the methods section. If you did this, and used the same data to measure these things, shouldn’t you correct the p-value here through a Bonferroni’s?
5) I’ve seen a slide in the first figure. Did you do slides as well? If yes, this should be in the methods section.
6) I’d say that it is most likely that the birds you captured where in a chronic phase of infection. How about getting examples showing no effects of parasites in chronic-infected birds? That will actually help you make the point here. Also, we tend to capture birds in the chronic phase of infection, because acute-infected birds have low mobility, as you said. Therefore, these would be a more plausible explanation for your results. Try to reorganize your discussion to make it clearer in this way.

Experimental design

Specific comments
Tittle
Line 2: I’d remove “the” the tittle, and place Bananaquit in plural.
Abstract
Lien 12:21: I think the introduction part of your abstract was very disproportioned.
Suggestion:
Parasite transmission is a heterogenous process in host-parasites interactions, being particularly important for vector-borne diseases. Haemosporidian parasites, a widespread protist, cause a malaria-like disease in birds and is often cause detrimental effects in hosts. In the Caribbeans, where malarial parasites are endemic, studying host-parasites interactions may give us important insights about energetic trade-offs involved in malarial parasites infections in birds.
Line 23: I wouldn’t use virulence here, but instead disease occurrence. You measured disease occurrence and not parasitemia here.
Line 26: I’d use Parahaemoproteus instead.
Line 31: I’d remove the word virulence since you studied occurrence of disease and not severity.
Line 32: Replace “malaria” with “malarial”.
Keywords
Line 33-34: Bananaquit, birds, body condition, and Haemoproteus are all in the tittle. I’d suggest placing other words to allow a greater generalization for your paper. Also, Haemoproteus should be in italics.
Introduction
Line 44: Fitness is a multidimensional word, so I’d use a different word here. What exactly do you mean by fitness reduction here?
Line 45: Cost to whom? The parasite or to the host?
Line 46: I’d remove “eco-evolutionary”.
Line 46 – 51: I think you could wrap up a little bit these sentences and create two ones. Suggestion:
Virulence has traditionally been associated to an unavoidable cost to hosts, since parasites reproduce at the expense of the host’s resources (REF). Because of the heterogeneity of parasite transmission, the relationship between the resistance of hosts and virulence of parasites may be highly dependent on the host immunity, as well as environmental conditions (REFs).
Line 52-53: Another suggestion:
Parasite transmission is particularly important in vector-born diseases because infection of susceptible hosts depends on infected vectors finding competent hosts (REF).
Line 55: Change “malaria” to “malarial”
Suggestion:
Haemosporidian parasites (Order Haemosporida, genera Plasmodium, Haemoproteus, Parahaemoproteus, and Leucocytozoon) are worldwide protists infecting birds of different families, causing a malaria-like disease (Valkiūnas, 2014).
Line 56: What do you mean by host fitness? Again, this is a multifunctional word.
Line 56-63:
Suggestion:
Malarial parasites may cause detrimental effects on hosts, such as increasing mortality, and decreasing overall body condition (REFs). On the other hand, particularly where haemosporidian parasites are endemic, there might not be any negative reported effects to the hosts (REFs). WRITE A COUPLE OF EXAMPLES HERE.
Line 61 – 65: I was uncertain here. Don’t you think that the hosts might be under a chronic phase of infection? It is very likely that we capture birds on the chronic phase of infection, and, therefore, the effects might not be that evident. Therefore, I suggest rephrasing this part of your paragraph. At least taking into account chronic infections here.
Line 66-69:
Suggestion:
Haemosporidian parasites may have different effects depending on the age and sex of individuals, such that juveniles tend to develop a more severe infection, and even have higher mortality compared to adults (REFs).
Line 69-72: You mean here that juveniles tend to be under a higher predation risk? Therefore, they may have higher corticosterone being produced. Is that what you meant? If yes, I suggest rephrasing the sentence and clearly pointing out at least one example linking predation risk, juveniles, and corticosterone. Also, I think this might have some room for uncertainty. Therefore, I suggest that you keep only the naïve immunity as a major cause.
Line 74: By higher virulence, do you mean a higher parasitemia? Also, I believe that you can add a sentence introducing the effect on sex first, instead of going directly to the example. Which sex tend to be more parasitized? The example here was relating infection, body condition, and sex. I believe it was a drastic transition.
Line 78: What do you mean by virulence here? Developing a more severe infection (a.k.a. parasitemia)?
Line 80: I believe you can start this paragraph with your objective directly.
Line 82: Change “malaria” to “malarial”
Line 80-87: I understand here that you are trying to explain to the reader why studying in the Caribbean is important. I would just keep it to a single sentence, though.
Line 88: Change “malaria” to “malarial”
Line 90: Change “ask” to the past, so that it reads “asked”. Also, I missed the prediction for your questions…
Suggestion:
We studied the relationship between infection by haemosporidian parasites, body condition, age, and sex in bananaquits (Coereba flaveola). We expected that infected individuals to have a lower body condition, and that juveniles to have a higher occurrence compared to adults. We were uncertain on what to expect for sex, because EXPLAIN YOUR REASONS…
Methods
Line 97: Change it to: we conducted the study from June 2018 to January 2019.
Line 98: What do you mean by visited 1 to 4 times? More explanation is needed here. I’d also remove the word “heavily”
Line 103: Are you specifically talking about Haemoproteus Parahaemoproteus and not Haemoproteus Haemoproteus, right? Some recent studies are already treating them as separate genera. I believe you could do the same here.
Line 104: Change authority to license.
Line 108: How many mist nets did you use? When did you open the nest? What time did you close the nets?
Line 108-109: How did you sex individuals? Only by the brood patch? If you said that they mostly breed from June to January, and you captured the individuals from January to June, how reliable would the brood patch be to sex individuals?
Line 110-111: I’d remove this sentence (saying about the difficulty in aging and sexing)
Line 129: If it is an unpublished data, I suggest writing the primer sequence here.
Line 124: How many times did you do a PCR for each individual? Once? If, yes, state in here.
Line 128: Where did you use BLAST? MalAvi or another site?
Line 130: Did you have a high incidence of unclassified? How did you deal with those?
Line 132: Did you remove those individuals? How many were left to analyze the Parahaemoproteus infections?
Line 142: did you mean “mass” when you said weight?
Line 142: You mentioned about the distance? Which distance was that? I’d just say “relationship” instead.
Line 146: You mentioned similar studies, in the plural, but only cite one. Maybe it would be good to cire more studies than.
Line 152: Change “see below”, to “see results section”
Line 158: For distribution assumption of normality, did you mean homoscedasticity?
Line 161-164: How did you compare the models? Did you use AIC or an automated process? Also, did you test for an interaction between age and sex?
Line 164-167: If you mentioned the packages here, they all have citations to be added as well.

Validity of the findings

Results
Line 172: What do you mean by “there was no blood”? I’d reword here. Did you try the other wing as well, whenever that happened?
Line 170-178: I’d just mention the total number of screened individuals. Ok, so you mentioned the number here… how many did you use in the statistical analysis? In age and sex, did you include unclassified in the models? I think more information on the actual number that you use in the models is needed here. Also, how many models did you run?
Line 180, 182: Strong relationship with a R2 of 40%and 13%? I’d say low to moderate on all of them, though. Were these metrics somehow correlated?
Line 184, 186: I’d place the mean and SE in parenthesis after the word “ones”.
Line 193: What does qqplot stand for? Write here, please.
Line 200-207: Do, you are using the same dataset to run 6 different models? Shouldn’t you correct the p-value here through a Bonferroni’s?
Discussion
Line 209: Can you cite those papers here?
Line 211: What do you mean by host fitness? This is a multifunctional word, please be specific.
Line 215: I’d be more careful with the word virulence. Just write detrimental or negative effects, instead. Since these results were only for Parahaemoproteus, and you did not present the lineage identity of the parasites, the idea of a parasite being virulent or not, is difficult to be inferred without doing an experimental approach here.
Line 220: Word “response” repeated two times in a roll. Can you replace the second one? You might need to rephrase the sentence.
Line 223: Can you show a different example here? Dragonflies might be a very distant group to make your point here. Also, were these dragonflies in a higher overall condition compared to uninfected ones?
Line 220-240: I feel like you are talking more about mass accumulation than the relationship between infection and body condition. I’d remove this paragraph. Also, you finished by saying that it may not be supported in your study system.
line 243: I’d be more specific when you mention the word fitness. What exactly are you referring to? This is a multifunctional word.
Line 246: This example is in survival. Do you have more examples on body condition itself?
Line 255-262: I’d say that it is most likely that the birds you captured where in a chronic phase of infection. How about getting examples showing no effects of parasites in chronic-infected birds? That will actually help you make the point here. Also, we tend to capture birds in the chronic phase of infection, because acute-infected birds have low mobility, as you said. Therefore, this would be a more plausible explanation for your results.
Line 264: How would you suggest people to test that, using specific metrics?
Line 265-272: Although this is an important study, I’d say that is both host and parasite specific. In some cases, it is true, infected organisms might forage more, and even be more exposed to predation (some small mammals), whereas some other groups infected with specific parasites might stay with a lower mobility. It would help if you got examples that are from groups closer to the birds you are studying.
Line 275-276: I think this sentence is a bit out of context here? General individuals from which group? Birds?
Line 280: Do these areas really have higher resource availability? If yes, cite the studies showing that.
Line 285: I feel like there should be more examples indicating this with birds and parasites. If you find more studies suggesting this pattern, than it would be a great fit.
Line 286-298: Indeed, that might be an important determinant. However, you could give the reader a sense of the mechanisms underlying this reason. After the hurricane, what happened with the resources? Were these infected birds with a lower body condition simply killed as a result of the hurricane or was it an indirect effect? Are there any studies indicating these mechanisms?
Line 299: What do you mean by virulence here? Did you mean parasite occurrence, which was what you measured?
Line 301: Virulence traits of whom? The hosts or the parasites? Be more specific.
Line 304-309: I wouldn’t account too much on the statistical power, instead try to find more studies that did not find any significant relationship between sex, age, and haemosporidian parasite occurrence. There are many.
Conclusions
Line 318: Why was your study valuable? By answering this question, you would make a more impacting sentence at the end to finish the manuscript up.

Additional comments

Figures
First figure:
Replace word “haemoparasite” to haemosporidian.
Second “haemoparasite”, change it to Parahaemoproteus. Did you do confirmatory slides? If yes, this should be in the methods section.
Second figure:
Change first “of” to between.
Change Haemoproteus to Parahaemoproteus.

---

## Round 0.3 · Minor Revisions

Dear Authors, One of the reviewers still has minor comments and suggested some changes in the current version of the manuscript, it would be great if you could address these.

Reviewer 1 ·

Basic reporting

Having already reviewed two previous versions of this paper, I feel that authors have suitably addressed all of my previous comments. This version is much-improved and I have no further suggestions for changes to the manuscript.

Experimental design

No comment

Validity of the findings

No comment

Additional comments

No comment

Reviewer 2 ·

Basic reporting

General comments
Gutiérrez-Ramos and Acevedo studied the relationship between body condition (three different indices), parasite occurrence, sex, and age in bananaquits. They found that infected individuals had a higher body condition compared to uninfected ones. The authors did a great job in answering all my previous questions, and now I only have some minor comments.
1) I still believe that you should better explain how you sexed individuals. Based on your explanations, you captured individuals in June/18, July/18, August/18, September/18, October/18, November/18, December/18, and January/18 and you said that bananaquits mostly breed in February, March, April, May, and June. Therefore, there is little to no overlap in the time where you could be more certain on the brood patch here. Therefore, I ask again, how reliable is the sexing procedure?
2) You mentioned that you did not account the Plasmodium-infected individuals? Does it mean that you remove these three individuals from statistical analysis? Please, clarify.
3) Based on how you showed your results, you ran six different models, being body weight / wing length ̴ infection occurrence, wing length / body weight ̴ infection occurrence, PCA ̴ infection occurrence, body weight / wing length ̴ sex * age, wing length / body weight ̴ sex * age, PCA ̴ sex * age. If you ran three models only and presented the results separately, you should specifically clarify this.
4) I was not very well convinced on the way you presented the explanation on virulence (rebuttal). To measure virulence, you would need to measure the immune response against the parasite, more appropriately. Therefore, you did not measure virulence, you measure parasite occurrence. You know if an individual is infected or not, you don’t know the parasitemia, or the differential immune response against the parasites within the bird hosts. Therefore, I’d be more careful in the terminology here.
5) In the discussion, you presented the metabolic syndrome hypothesis. However, you said that this was unlikely the case in your study. If you believe that this is not the case, why mention it at all? Afterwards, you stated that immunological differences may lead to parasite resistance or host tolerance. However, I felt that there are some loose ends. How do you get from having a more resistance against a parasite or being more tolerant and having a higher body condition with a haemosporidian infection? The way you presented, it looks like that only resistant or tolerant individuals were infected, and that is why they may have a higher body condition, which I don’t believe it is the case. Please, make this connection a bit clearer, to help the reader identify how immunology, parasitism, and overall health status are connected.
6) Lastly, you presented the important environmental factor, which was the hurricane. However, I believe that the way you presented, I got the impression that you suggested that only “high quality” individuals survived, and because of that they may have a higher body condition, even when infected. However, based on this reasoning, how can you explain that the uninfected “high-quality” individuals (because they survived the hurricane) had a lower body condition? Therefore, I don’t think this example helped much in understanding this situation, unless you try to reason it with a different perspective.
Congratulations on tidying the manuscript, and please check some specific comments below.

Experimental design

Specific comments
Abstract
Line 14: Replace “heterogeneity” with “complexity”.

Introduction
Line 31: Add “of” after consequence.
Line 32: Remove “the” before parasite, make “parasite” in plural, and replace “reproduces” with “reproduce”.
Line 38: What do you mean by “host virulence”? Did you mean virulence from parasites instead? Or consequences of the parasitism to hosts? Clarify, please.
Line 39-41: I believe that you could rephrase this sentence. At the current stage, what you wrote conveys the information that vectors intentionally try to find competent hosts, which may not be the case. If the vector is not specialist, it may utilize resources from competent and non-competent hosts, as described by the dilution effect.
Line 43: In your rebuttal letter you said that you would be conservative and include only Haemoproteus. If that is still the case, I suggest you remove Parahaemoproteus from this sentence.
Line 44: Add ‘s after “bird”.
Line 60: Replace the words “higher virulence” with “a higher parasite load”.
Line 63: Remove “Tawny pipits” and make “male” in the plural form.
Line 54: Replace “this pattern” with “the”.
Line 65: Replace “is” with “are” if you make the change on line 64.
Line 65: Put “shows” in simple past form.
Line 66: This study showed no clear effect of the parasitism on the viability comparing males and females. I suggest rephrasing this sentence.
Line 73: What do you mean by “mediated by sex”? Clarify, please.
Line 75: What do you mean by “recent syntheses”? Clarify, please.
Line 76.77: I believe this is a very broad sentence to be placed as the last sentence. I suggest removing it.
Methods
Line 81: Replace “malaria” with “malarial”.
Line 84: Replace “malaria” with “malarial”.
Line 93: Put “show” in the simple past.
Line 112: Put a comma after “vein”.
Line 115: You performed the PCR to screen for the parasite presence, not sequence, right? Later, I assume, you tried to sequence the ones that were positive. Please clarify, please.
Line 116: Since this is an unpublished primer, I believe you could place the base pairs here.
Line 129: Did you remove these three Plasmodium-infected individuals?
Line 144: Replace “malaria” with “malarial”.
Results
Line 180: Since you mentioned this in the methods, I suggest removing this from here “Plasmodium infected individuals…”.

Validity of the findings

Discussion
Line 212: Replace “malaria” with “malarial”.
Line 213: Replace “less” with “little”.
Line 224: Add a comma after “blood”.
Line 226: Replace “better” with “higher”.
Line 231: Add citation after “forests”.
Line 237: Add “-“ after “sub”.
Line 254: Remove the extra “.”.
Line 272: Remove “a” before “generalist” and add “and” after “generalist”.
Line 289: You did not measure consequences of infection, but parasite occurrence differences between males and females.
Line 299: Replace “effect” with difference”. Remove “.” before “but”, and add a “comma” with a lowercase “but”.

Additional comments

NA

---

## Round 0.4 · accepted · Accept

The authors have addressed most of the comments and concerns raised by the reviewers. I am happy with the revised version of the manuscript and hence I accept the manuscript for publication.